# On Adversarial Mixup Resynthesis

**Christopher Beckham**[1,3]**, Sina Honari**[1,3]**, Vikas Verma**[1,6,†]**, Alex Lamb**[1,2]**, Farnoosh Ghadiri**[1,3]**,
R Devon Hjelm**[1,2,5]**, Yoshua Bengio**[1,2,*]** & Christopher Pal**[1,3,4,‡,*]**
[1]Mila - Québec Artificial Intelligence Institute, Montréal, Canada
[2]Université de Montréal, Canada
[3]Polytechnique Montréal, Canada
[4]Element AI, Montréal, Canada
[5]Microsoft Research, Montréal, Canada
[6]Aalto University, Finland
firstname.lastname@mila.quebec
[†] vikas.verma@aalto.fi, [‡] christopher.pal@polymtl.ca

## Abstract

In this paper, we explore new approaches to combining information encoded within the learned representations of auto-encoders. We explore models that are capable of combining the attributes of multiple inputs such that a resynthesised output is trained to fool an adversarial discriminator for real versus synthesised data. Furthermore, we explore the use of such an architecture in the context of semi-supervised learning, where we learn a mixing function whose objective is to produce interpolations of hidden states, or masked combinations of latent representations that are consistent with a conditioned class label. We show quantitative and qualitative evidence that such a formulation is an interesting avenue of research.[1]

## 1 Introduction

The auto-encoder is a fundamental building block in unsupervised learning. Auto-encoders are trained to reconstruct their inputs after being processed by two neural networks: an encoder which encodes the input to a high-level representation or *bottleneck*, and a decoder which performs the reconstruction using that representation as input. One primary goal of the auto-encoder is to learn representations of the input data which are useful (Bengio, 2012), which may help in downstream tasks such as classification (Zhang et al., 2017; Hsu et al., 2019) or reinforcement learning (van den Oord et al., 2017; Ha & Schmidhuber, 2018). The representations of auto-encoders can be encouraged to contain more 'useful' information by restricting the size of the bottleneck, through the use of input noise (e.g., in denoising auto-encoders, Vincent et al., 2008), through regularisation of the encoder function (Rifai et al., 2011), or by introducing a prior (Kingma & Welling, 2013). Other goals include learning interpretable representations (Chen et al., 2016; Jang et al., 2016), disentanglement of latent variables (Liu et al., 2017; Thomas et al., 2017) or maximisation of mutual information (Chen et al., 2016; Belghazi et al., 2018; Hjelm et al., 2019) between the input and the code.

We know that data augmentation greatly helps when it comes to increasing generalisation performance of models. A practical intuition for why this is the case is that by generating additional samples, we are training our model on a set of examples that better covers those in the test set. In the case of images, we are already afforded a variety of transformation techniques at our disposal, such as random flipping, crops, rotations, and colour jitter. While indispensible, there are other regularisation techniques one can also consider.

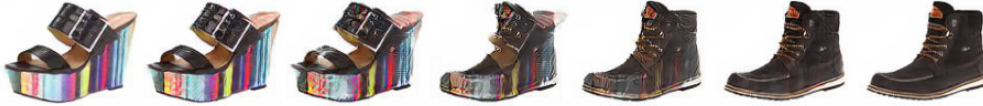

Figure 1: Adversarial mixup resynthesis involves mixing the latent codes used by auto-encoders through an arbitrary mixing mechanism that is able to recombine codes from different inputs to produce novel examples. These novel examples are made to look realistic via the use of adversarial learning. We show the gradual mixing between two real examples of shoes (far left and far right).

*Mixup* (Zhang et al., 2018) is a regularisation technique which encourages deep neural networks to behave linearly between pairs of data points. These methods artificially augment the training set by producing random convex combinations between pairs of examples and their corresponding labels and training the network on these combinations. This has the effect of creating smoother decision boundaries, which was shown to have a positive effect on generalisation performance. Arguably however, the downside of *mixup* is that these random convex combinations between images may not look realistic due to the interpolations being performed on a per-pixel level.

In Verma et al. (2018); Yaguchi et al. (2019), these random convex combinations are computed in the *hidden space* of the network. This procedure can be viewed as using the high-level representation of the network to produce novel training examples. Though mixing based methods have shown to improve strong baselines in supervised learning (Zhang et al., 2018; Verma et al., 2018) and semi-supervised learning (Verma et al., 2019a; Berthelot et al., 2019; Verma et al., 2019b), there has been relatively less exploration of these methods in the context of unsupervised learning.

This kind of mixing (in latent space) may encourage representations which are more amenable to the idea of *systematic generalisation* – we would like our model to be able to compose new examples from unseen combinations of latent factors despite only seeing a very small subset of those combinations in training (Bahdanau et al., 2018). Therefore, in this paper we explore the use of such a mechanism in the context of auto-encoders through an exploration of various *mixing functions*. These mixing functions could consist of continuous interpolations between latent vectors such as in Verma et al. (2018), genetically-inspired recombination such as crossover, or even a deep neural network which learns the mixing operation. To ensure that the output of the decoder given the mixed representation resembles the data distribution at the pixel level, we leverage adversarial learning (Goodfellow et al., 2014), where here we train a discriminator to distinguish between decoded mixed and real data points. This gives us the ability to simulate novel data points (through *exponentially many* combinations of latent factors not present in the training set), and also improve the learned representation as we will demonstrate on downstream tasks later in this paper. Figure 1 shows one example of such mixing.

## 2 Formulation

The auto-encoder serves as the baseline for our work since its encoder allows us to infer latent variables, and therefore also allow us to compute mixing operations between those variables. Subsequently, the decoder allows us to visualise these mixed latent variables and (through an adversarial framework) enable us to leverage those mixes to improve representations learned by the auto-encoder. Let us consider an auto-encoder model $F(\cdot)$, with the encoder part denoted as $f(\cdot)$ and the decoder $g(\cdot)$. In an auto-encoder we wish to minimise the reconstruction, which is simply:

$$\min_F \mathbb{E}_{\mathbf{x}\sim\mathbf{p}(\mathbf{x})} \|\mathbf{x} - g(f(\mathbf{x}))\|_2 \tag{1}$$

Because auto-encoders that are trained by pixel-space reconstruction produce low quality images (characterized by blurriness), we augment this baseline by adding an adversarial game to the reconstruction (as done in Larsen et al. (2016)). In turn, the discriminator $D$ tries to distinguish between real and reconstructed $\mathbf{x}$, and the auto-encoder tries to construct 'realistic' reconstructions so as to fool the discriminator. This formulation serves as our *baseline* (to make this clear throughout this work, we call this 'AE + GAN'), which can be written as:

$$\min_F \ \mathbb{E}_{\mathbf{x}\sim\mathbf{p}(\mathbf{x})} \lambda \|\mathbf{x} - g(f(\mathbf{x}))\|_2 + \ell_{GAN}(D(g(f(\mathbf{x}))), 1)$$
$$\min_D \ \mathbb{E}_{\mathbf{x}\sim\mathbf{p}(\mathbf{x})} \ell_{GAN}(D(\mathbf{x}), 1) + \ell_{GAN}(D(g(f(\mathbf{x}))), 0), \tag{2}$$

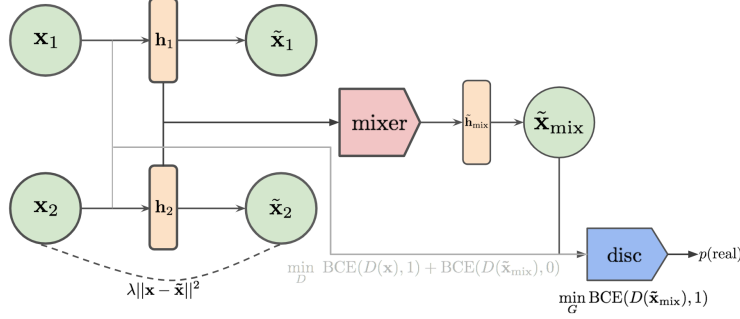

Figure 2: The unsupervised version of adversarial mixup resynthesis (AMR). In addition to the auto-encoder loss functions, we have a mixing function Mix (called 'mixer' in the figure) which creates some combination between the latent variables $\mathbf{h}_1$ and $\mathbf{h}_2$, which is subsequently decoded into an image intended to be realistic-looking by fooling the discriminator. Subsequently the discriminator's job is to distinguish real samples from generated ones from mixes.

where $\ell_{GAN}$ is a GAN-specific loss function. In our case, $\ell_{GAN}$ is the binary cross-entropy loss, which corresponds to the Jenson-Shannon GAN (Goodfellow et al., 2014).

What we would like to do is to be able to encode an arbitrary pair of inputs $\mathbf{h}_1 = f(\mathbf{x_1})$ and $\mathbf{h}_2 = f(\mathbf{x_2})$ into their latent representation, perform some combination between them through a function we denote $\text{Mix}(\mathbf{h_1}, \mathbf{h_2})$ (more on this soon), run the result through the decoder $\mathbf{g}(\cdot)$, and then minimise some loss function which encourages the resulting decoded mix to look realistic. With this in mind, we propose *adversarial mixup resynthesis* (AMR), where part of the auto-encoder's objective is to produce mixes which, when decoded, are indistinguishable from real images. The generator and the discriminator of AMR are trained by the following mixture of loss components:

$$\min_F \mathbb{E}_{\mathbf{x},\mathbf{x}'\sim\mathbf{p}(\mathbf{x})} \underbrace{\lambda \left\|\mathbf{x} - g(f(\mathbf{x}))\right\|_2}_{\text{reconstruction}} + \underbrace{\ell_{GAN}(D(g(f(\mathbf{x}))), 1)}_{\text{fool D with reconstruction}} + \underbrace{\ell_{GAN}(D(g(\text{Mix}(f(\mathbf{x}), f(\mathbf{x}')))), 1)}_{\text{fool D with mixes}}$$

$$\min_D \mathbb{E}_{\mathbf{x},\mathbf{x}'\sim\mathbf{p}(\mathbf{x})} \underbrace{\ell_{GAN}(D(\mathbf{x}), 1)}_{\text{label x as real}} + \underbrace{\ell_{GAN}(D(g(f(\mathbf{x}))), 0)}_{\text{label reconstruction as fake}} + \underbrace{\ell_{GAN}(D(g(\text{Mix}(f(\mathbf{x}), f(\mathbf{x}')))), 0)}_{\text{label mixes as fake}}.$$

(3)

The AMR model is shown in Figure 2. There are many ways one could combine the two latent representations, and we denote this function $\text{Mix}(\mathbf{h_1}, \mathbf{h_2})$. Manifold mixup (Verma et al., 2018) implements mixing in the hidden space through convex combinations:

$$\text{Mix}_{\text{mixup}}(\mathbf{h}_1, \mathbf{h}_2) = \alpha \mathbf{h}_1 + (1 - \alpha)\mathbf{h}_2, \tag{4}$$

where $\alpha \in [0, 1]$ is sampled from a Uniform$(0, 1)$ distribution. We can interpret this as interpolating along *line segments*, as shown in Figure 3 (left).

We also explore a strategy in which we randomly retain some components of the hidden representation from $\mathbf{h}_1$ and use the rest from $\mathbf{h}_2$. In this case we would randomly sample a binary mask $\mathbf{m} \in \{0, 1\}^k$ (where $k$ denotes the number of feature maps) and perform the following operation:

$$\text{Mix}_{\text{Bern}}(\mathbf{h}_1, \mathbf{h}_2) = \mathbf{m}\mathbf{h}_1 + (1 - \mathbf{m})\mathbf{h}_2, \tag{5}$$

where $\mathbf{m}$ is sampled from a Bernoulli$(p)$ distribution ($p$ can simply be sampled uniformly) and multiplication is element-wise. This formulation is interesting in the sense that it is very reminiscent of crossover in biological reproduction: the auto-encoder has to organise feature maps in such a way that that *any* recombination between sets of feature maps must decode into realistic looking images.

## 2.1 Mixing with k examples

We can generalise the above mixing functions to operate on more than just two examples. For instance, in the case of mixup (Equation 4), if we were to mix between examples $\{\mathbf{h}_1, \ldots, \mathbf{h}_k\}$, we

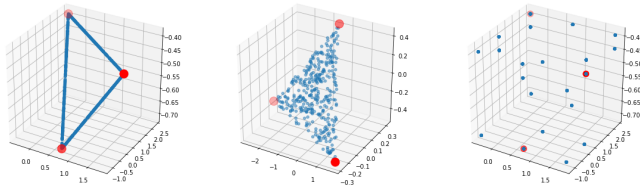

Figure 3: Left: mixup (Equation 4), with interpolated points in blue corresponding to line segments between the three points shown in red. Middle: triplet mixup (Equation 6). Right: Bernoulli mixup (Equation 5).

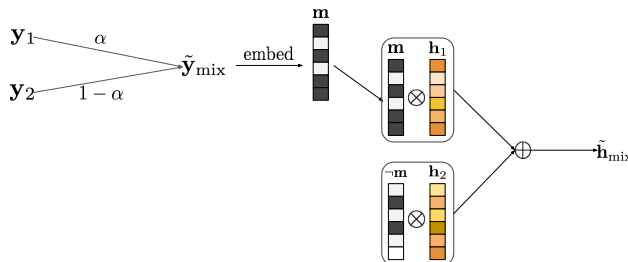

Figure 4: The supervised version of Bernoulli mixup. In this, we learn an embedding function embed($\mathbf{y}$) (an MLP) which maps $\mathbf{y}$ to Bernoulli parameters $\mathbf{p} \in [0,1]^k$, from which a Bernoulli mask $\mathbf{m} \sim$ Bernoulli($\mathbf{p}$) is sampled. The resulting mix is then simply $\mathbf{mh}_1 + (1 - \mathbf{m})\mathbf{h}_2$. Intuitively, the embedding function can be thought of as a function which decides what feature maps need to be recombined from $\mathbf{h}_1$ and $\mathbf{h}_2$ in order to produce a mix which satisfies the attribute vector $\mathbf{y}$.

can simply sample $\boldsymbol{\alpha} \sim$ Dirichlet$(1, \ldots, 1)^2$, where $\boldsymbol{\alpha} \in [0,1]^k$ and $\sum_{i=1}^{k} \alpha_i = 1$ and compute the dot product between this and the hidden states:

$$\alpha_1 \cdot \mathbf{h}_1 + \cdots + \alpha_k \cdot \mathbf{h}_k = \sum_{j=1}^{k} \alpha_j \mathbf{h}_j, \tag{6}$$

One can think of this process as being equivalent to doing multiple iterations (or in biological terms, generations) of mixing. For example, in the case of a large $k$, $\alpha_1 \cdot \mathbf{h}_1 + \alpha_2 \cdot \mathbf{h}_2 + \alpha_3 \cdot \mathbf{h}_3 + \cdots = \underbrace{(\ldots \underbrace{(\alpha_1 \cdot \mathbf{h}_1 + \alpha_2 \cdot \mathbf{h}_2)}_{\text{first iteration}} + \mathbf{h}_3 \cdot \alpha_3) + \ldots}_{\text{second iteration}}$. We show the $k = 3$ case in in Figure 3 (middle).

## 2.2 Using labels

While it is interesting to generate new examples via random mixing strategies in the hidden states, we also explore a supervised formulation in which we learn a function that can produce *specific kinds* of mixes between two examples such that they are consistent with a particular class label. We make this possible by backpropagating through a classifier network $p(\mathbf{y}|\mathbf{x})$ which branches off the end of the discriminator, i.e., an auxiliary classifier GAN (Odena et al., 2017).

Let us assume that for some image $\mathbf{x}$, we have a set of associated binary attributes $\mathbf{y}$, where $\mathbf{y} \in \{0,1\}^k$ (and $k \geq 1$). We introduce an embedding function embed($\mathbf{y}$), which is an MLP (whose parameters are learned in unison with the auto-encoder) that maps $\mathbf{y}$ to Bernoulli parameters $\mathbf{p} \in [0,1]^k$. These parameters are used to sample a Bernoulli mask $\mathbf{m} \sim$ Bernoulli($\mathbf{p}$) to produce a

new combination trained to have the class label $\mathbf{y}$ (for the sake of convenience, we can summarize the embedding and sampling steps as simply $\mathrm{Mix_{sup}}(\mathbf{h}_1, \mathbf{h}_2, \mathbf{y})$). Note that the conditioning class label should be semantically meaningful with respect to both of the conditioned hidden states. For example, if we're producing mixes based on the gender attribute and both $\mathbf{h}_1$ and $\mathbf{h}_2$ are male, it would not make sense to condition on the 'female' label since the class mixer only recombines rather than adding new information. To enforce this constraint, during training we simply make the conditioning label a convex combination $\tilde{\mathbf{y}}_{\mathrm{mix}} = \alpha \mathbf{y}_1 + (1 - \alpha)\mathbf{y}_2$ as well, using $\alpha \sim \mathrm{Uniform}(0, 1)$. This is summarised in Figure 4.

Concretely, the auto-encoder and discriminator, in addition to their unsupervised losses described in Equation 3, try to minimise their respective supervised losses:

$$\min_F \ \mathbb{E}_{\mathbf{x}_1,\mathbf{y}_1 \sim p(\mathbf{x},\mathbf{y}), \mathbf{x}_2,\mathbf{y}_2 \sim p(\mathbf{x},\mathbf{y}), \alpha \sim U(0,1)} \ \underbrace{\ell_{\mathrm{GAN}}(D(g(\tilde{\mathbf{h}}_{\mathrm{mix}})), 1)}_{\text{fool } D \text{ with mix}} + \underbrace{\ell_{\mathrm{cls}}(p(\mathbf{y}|\mathbf{g}(\tilde{\mathbf{h}}_{\mathrm{mix}})), \tilde{\mathbf{y}}_{\mathrm{mix}})}_{\text{make mix's class consistent}}$$

$$\min_D \ \mathbb{E}_{\mathbf{x}_1,\mathbf{y}_2 \sim p(\mathbf{x},\mathbf{y}), \mathbf{x}_2,\mathbf{y}_2 \sim p(\mathbf{x},\mathbf{y}), \alpha \sim U(0,1)} \ \underbrace{\ell_{\mathrm{GAN}}(D(g(\tilde{\mathbf{h}}_{\mathrm{mix}})), 0)}_{\text{label mixes as fake}} \quad (7)$$

$$\text{where } \tilde{\mathbf{y}}_{\mathrm{mix}} = \alpha \mathbf{y}_1 + (1 - \alpha)\mathbf{y}_2 \text{ and } \tilde{\mathbf{h}}_{\mathrm{mix}} = \mathrm{Mix_{sup}}(f(\mathbf{x}_1), f(\mathbf{x}_2), \tilde{\mathbf{y}}_{\mathrm{mix}})$$

## 3 Related work

Our method can be thought of as an extension of auto-encoders that allows for sampling through mixing operations, such as continuous interpolations and masking operations. Variational auto-encoders (VAEs, Kingma & Welling, 2013) can also be thought of as a similar extension of auto-encoders, using the outputs of the encoder as parameters for an approximate posterior $q(\mathbf{z}|\mathbf{x})$ which is matched to a prior distribution $p(\mathbf{z})$ through the evidence lower bound objective (ELBO). At test time, new data points are sampled by passing samples from the prior, $\mathbf{z} \sim p(\mathbf{z})$, through the decoder. The fundamental difference here is that the output of the encoder is constrained to come from a pre-defined prior distribution, whereas we impose no constraint, at least not in the probabilistic sense.

The ACAI algorithm (adversarially constrained auto-encoder interpolation) is another approach which involves sampling interpolations as part of an unsupervised objective (Berthelot et al., 2019). ACAI uses a discriminator network to predict the mixing coefficient $\alpha$ from the decoded output of the mixed representation, and the auto-encoder tries to 'fool' the discriminator by making it predict either $\alpha = 0$ or $\alpha = 1$, making interpolated points indistinguishable from real ones. One of the main differences is that in our framework the discriminator output is agnostic to the mixing function used, so rather than trying to predict the parameter(s) of the mix (in this case, $\alpha$) it is only required to predict whether the mix is real or fake (1/0). On a more technical level, the type of GAN they employ is the least squares GAN (Mao et al., 2017), whereas we use JSGAN (Goodfellow et al., 2014) and spectral normalization (Miyato et al., 2018) to impose a Lipschitz constraint on the discriminator, which is known to be very effective in minimising stability issues in training.

The GAIA algorithm (Sainburg et al., 2018) uses a BEGAN framework with an additional interpolation-based adversarial objective. In this work, the mixing function involves interpolating with an $\alpha \sim \mathcal{N}(\mu, \sigma)$, where $\mu$ is defined as the midpoint between the two hidden states $\mathbf{h}_1$ and $\mathbf{h}_2$. For their supervised formulation, the authors use a simple technique in which average latent vectors are computed over images with particular attributes. For example, $\bar{\mathbf{h}}_{\mathrm{female}}$ and $\bar{\mathbf{h}}_{\mathrm{glasses}}$ could denote the average latent vectors over all images of women and all images of people wearing glasses, respectively. One can then perform arithmetic over these different vectors to produce novel images, e.g. $\bar{\mathbf{h}}_{\mathrm{female}} + \bar{\mathbf{h}}_{\mathrm{glasses}}$. However, this approach is crude in the sense that these vectors are confounded by and correlated with other irrelevant attributes in the dataset. Conversely, in our technique, we *learn* a mixing function which tries to produce combinations between latent states consistent with a class label by backpropagating through the classifier branch of the discriminator. If the resulting mix contains confounding attributes, then the mixing function would be penalised for doing so.

What primarily differentiates our work from theirs is that we perform an exploration into different kinds of mixing functions, including a semi-supervised variant which uses an MLP to produce mixes consistent with a class label. In addition to systematic generalisation, our work is partly motivated by processes which occur in sexual reproduction; for example, Bernoulli mixup can be seen as the analogue to crossover in the genetic algorithm setting, similar to how dropout (Srivastava et al., 2014) can be seen as being analogous to random mutations. We find this connection to be appealing, as

there has been some interest in leveraging concepts from evolution and biology in deep learning, for instance meta-learning (Bengio et al., 1991), dropout (as previously mentioned), biologically plausible deep learning (Bengio et al., 2015) and evolutionary strategies for reinforcement learning (Such et al., 2017; Salimans et al., 2017).

## 4 Results

In this section we evaluate the classification accuracy of AMR on various datasetss by training a linear classifier on the latent features of the unsupervised variant of the model. We also measure evaluate our model on a disentanglement task, which is also unsupervised. Finally, we demonstrate some qualitative results.

### 4.1 Classification of learned features

One way to evaluate the usefulness of the representation learned is to evaluate its performance on some downstream tasks. Similar to what was done in ACAI, we modify our training procedure by attaching a linear classification network to the output of the encoder and train it in unison with the other objectives. The classifier does not contribute any gradients back into the auto-encoder, so it simply acts as a probe (Alain & Bengio, 2016) whose accuracy can be monitored over time to quantify the usefulness of the representation learned by the encoder.

We employ the following datasets for classification: MNIST (Deng, 2012), KMNIST (Clanuwat et al., 2018), and SVHN (Netzer et al., 2011). We perform three runs for each experiment, and from each run we collect the highest accuracy on the validation set over the entire course of training, from which we compute the mean and standard deviation. Hyperparameter tuning on $\lambda$ was performed manually (this essentially controls the trade-off between the reconstruction and adversarial losses), and we experimented with a reasonable range of values (i.e. $\{2, 5, 10, 20, 50\}$). We experiment with three mixing functions: mixup (Equation 4), Bernoulli mixup (Equation 5)[3], and the various higher-order versions with $k > 2$ (see Section 2.1). The number of epochs we trained for is dependent on the dataset (since some datasets converged faster than others) and we indicate this in each table's caption.

In Table 1 we show results on relatively simple datasets – MNIST, KMNIST, and SVHN – with an encoding dimension of $d_h = 32$ (more concretely, a bottleneck of two feature maps of spatial dimension $4 \times 4$). In Table 2 we explore the effect of data ablation on SVHN with the same encoding dimension but randomly retaining 1k, 5k, 10k, and 20k examples in the training set, to examine the efficacy of AMR in the low-data setting. Lastly, in Table 3 we evaluate AMR in a higher dimensional setting, trying out SVHN with $d_h = 256$ (i.e., a spatial dimension of $16 \times 4 \times 4$) and CIFAR10 with $d_h = 256$ and $d_h = 1024$ (a spatial dimension of $64 \times 4 \times 4$). These encoding dimensions were chosen so as to conform to ACAI's experimental setup.

In terms of training hyperparameters, we used ADAM (Kingma & Ba, 2014) with a learning rate of $10^{-4}$, $\beta_1 = 0.5$ and $\beta_2 = 0.99$ and an L2 weight decay of $10^{-5}$. For architectural details, please consult the README file in the code repository.[4]

### 4.2 Disentanglement

Lastly, we run experiments on the DSprite (Matthey et al., 2017) dataset, a 2D sprite dataset whose images are generated with six known (ground truth) latent factors. Latent encodings produced by autoencoders trained on this dataset can be used in conjunction a disentanglement metric (see Higgins et al. (2017); Kim & Mnih (2018)), which measures the extent to which the learned encodings are able to recover the ground truth latent factors. These results are shown in Table 4. We can see that for the AMR methods, Bernoulli mixing performs the best, especially the triplet formulation. $\beta$-VAE performs the best overall, and this may be in part due to the fact that the prior distribution on the latent encoding is an independent Gaussian, which may encourage those variables to behave more independently.

Table 1: Classification accuracy results when training a linear classifier probe on top of the auto-encoder's encoder output ($d_h = 32$). Each experiment was run thrice. ($^\dagger$ = results taken from the original paper). MNIST, KMNIST, and SVHN were trained for 2k, 5k, and 4.5k epochs, respectively. AE+GAN = adversarial reconstruction auto-encoder (Equation 2); AMR = adversarial mixup resynthesis (ours); ACAI = adversarially constrained auto-encoder interpolation (Berthelot et al., 2019))

| Method | Mix | $k$ | MNIST | ($\lambda$) | KMNIST | ($\lambda$) | SVHN | ($\lambda$) |
|---|---|---|---|---|---|---|---|---|
| AE+GAN | - | - | $97.52 \pm 0.29$ | (5) | $76.18 \pm 1.79$ | (10) | $37.01 \pm 2.22$ | (5) |
| AMR | mixup | 2 | $98.01 \pm 0.10$ | (10) | $80.39 \pm 3.11$ | (10) | $43.98 \pm 3.05$ | (10) |
| | Bern | 2 | $97.76 \pm 0.58$ | (10) | $81.54 \pm 3.46$ | (10) | $38.31 \pm 2.68$ | (10) |
| | mixup | 3 | $97.61 \pm 0.15$ | (20) | $77.20 \pm 0.43$ | (10) | $\mathbf{47.34 \pm 3.79}$ | (10) |
| ACAI | mixup | 2 | $\mathbf{98.66 \pm 0.36}$ | (2) | $\mathbf{84.67 \pm 1.16}$ | (10) | $34.74 \pm 1.12$ | (2) |
| ACAI$^\dagger$ | mixup | 2 | $98.25 \pm 0.11$ | (N/A) | - | (N/A) | $34.47 \pm 1.14$ | (N/A) |

Table 2: Classification accuracy results when training a linear classifier probe on top of the auto-encoder's encoder output ($d_h = 32$) for various training set sizes for SVHN (1k, 5k, 10k, and 20k, for 6k, 6k, 6k, and 4k epochs respectively).

| Method | Mix | $k$ | SVHN(1k) | ($\lambda$) | SVHN(5k) | ($\lambda$) | SVHN(10k) | ($\lambda$) | SVHN(20k) | ($\lambda$) |
|---|---|---|---|---|---|---|---|---|---|---|
| AE+GAN | - | - | $22.71 \pm 0.73$ | (10) | $25.35 \pm 0.44$ | (10) | $26.18 \pm 0.81$ | (10) | $29.21 \pm 1.01$ | (20) |
| AMR | mixup | 2 | $21.89 \pm 0.19$ | (10) | $25.41 \pm 1.15$ | (20) | $30.87 \pm 0.74$ | (10) | $36.27 \pm 3.76$ | (10) |
| | Bern | 2 | $22.59 \pm 1.31$ | (20) | $26.07 \pm 1.87$ | (20) | $30.12 \pm 2.37$ | (10) | $35.98 \pm 0.56$ | (10) |
| | mixup | 3 | $22.96 \pm 0.69$ | (10) | $\mathbf{29.92 \pm 3.37}$ | (10) | $\mathbf{31.87 \pm 0.68}$ | (10) | $\mathbf{37.04 \pm 2.32}$ | (10) |
| ACAI | mixup | 2 | $\mathbf{24.15 \pm 1.65}$ | (10) | $29.58 \pm 1.08$ | (10) | $29.56 \pm 0.97$ | (2) | $31.23 \pm 0.31$ | (5) |

Table 3: Classification accuracy results on SVHN ($d_h = 256$) and CIFAR10 ($d_h \in \{256, 1024\}$). These configurations were trained for 4k, 3k, and 8k epochs, respectively. ($^\dagger$ = results from original paper.)

| Method | Mix | $k$ | SVHN (256) | ($\lambda$) | CIFAR10 (256) | ($\lambda$) | CIFAR10 (1024) | ($\lambda$) |
|---|---|---|---|---|---|---|---|---|
| AE+GAN | - | - | $59.00 \pm 0.12$ | (5) | $53.08 \pm 0.28$ | (50) | $59.93 \pm 0.60$ | (50) |
| AMR | mixup | 2 | $71.51 \pm 1.35$ | (5) | $54.24 \pm 0.42$ | (50) | $60.80 \pm 0.79$ | (50) |
| | Bern | 2 | $58.64 \pm 2.18$ | (10) | $52.40 \pm 0.51$ | (50) | $59.81 \pm 0.56$ | (50) |
| | mixup | 3 | $73.33 \pm 3.23$ | (5) | $\mathbf{54.94 \pm 0.37}$ | (50) | $61.68 \pm 0.67$ | (50) |
| | mixup | 4 | $74.69 \pm 1.11$ | (5) | $54.68 \pm 0.33$ | (50) | $\mathbf{61.72 \pm 0.20}$ | (50) |
| | mixup | 6 | $73.85 \pm 0.84$ | (5) | $52.95 \pm 0.92$ | (50) | $60.34 \pm 0.82$ | (50) |
| | mixup | 8 | $\mathbf{75.71 \pm 1.29}$ | (5) | $53.07 \pm 1.04$ | (50) | $59.75 \pm 1.04$ | (50) |
| ACAI | mixup | 2 | $68.64 \pm 1.50$ | (2) | $50.06 \pm 1.33$ | (20) | $57.42 \pm 1.29$ | (20) |
| ACAI$^\dagger$ | mixup | 2 | $85.14 \pm 0.20$ | (N/A) | $52.77 \pm 0.45$ | (N/A) | $63.99 \pm 0.47$ | (N/A) |

## 4.3 Qualitative results (unsupervised)

Due to space constraints, we show qualitative results in the supplementary material. We compare interpolations (between our technique, ACAI, AE+GAN, and pixel-space interpolation) on three datasets: SVHN (Netzer et al., 2011), CelebA (Liu et al., 2015), and Zappos shoes (Yu & Grauman, 2014, 2017). It can be easily seen that AMR produces realistic-looking mixes with significantly less 'ghosting' or 'artifacting' as exhibited in the baselines. This supplementary also explains an extra 'consistency loss' term which was used to improve the quality of the interpolation trajectory between two images.

Table 4: Results on DSprite using the disentanglement metric proposed in Kim & Mnih (2018). For $\beta$-VAE (Higgins et al., 2017), we show the results corresponding to the best-performing $\beta$ values. For AMR, $\lambda = 1$ since this performed the best.

| Method | Mix | $k$ | Accuracy |
|---|---|---|---|
| VAE($\beta = 100$) | - | - | **68.00 $\pm$ 3.89** |
| AE+GAN | - | - | 45.12 $\pm$ 2.68 |
| AMR | mixup | 2 | 49.00 $\pm$ 6.72 |
| | Bern | 2 | 53.00 $\pm$ 1.59 |
| | mixup | 3 | 51.13 $\pm$ 4.95 |
| | Bern | 3 | 56.00 $\pm$ 0.91 |

## 4.4 Qualitative results (supervised)

We present some qualitative results with the supervised formulation. We train our supervised AMR variant using a subset of the attributes in CelebA ('is male', 'is wearing heavy makeup', and 'is wearing lipstick'). We consider pairs of examples $\{(\mathbf{x}_1, \mathbf{y}_1), (\mathbf{x}_2, \mathbf{y}_2)\}$ (where one example is male and the other female) and produce random convex combinations of the attributes $\tilde{\mathbf{y}}_{\text{mix}} = \alpha \mathbf{y}_1 + (1 - \alpha)\mathbf{y}_2$ and decode their resulting mixes $\text{Mix}_{\text{sup}}(f(\mathbf{x}_1), f(\mathbf{x}_2), \tilde{\mathbf{y}}_{\text{mix}})$. This can be seen in Figure 5.

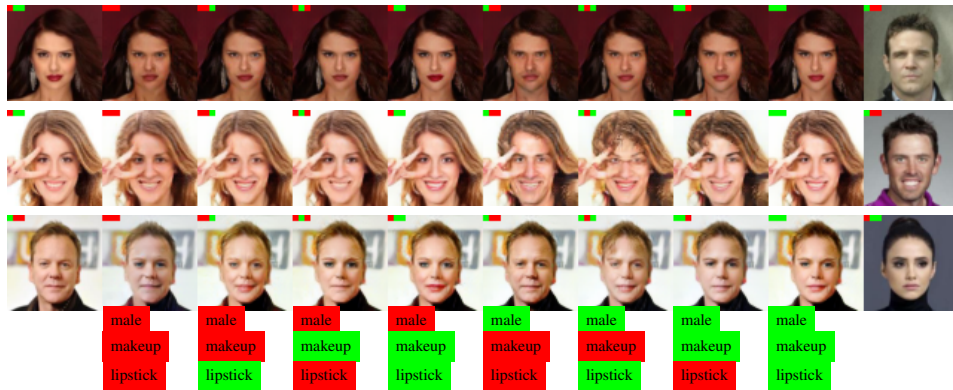

Figure 5: Interpolations produced by the class mixer function for the set of binary attributes {male, heavy makeup, lipstick}. For each image, the left-most face is $\mathbf{x}_1$ and the right-most face $\mathbf{x}_2$, with faces in between consisting of mixes $\text{Mix}_{\text{sup}}(f(\mathbf{x}_1), f(\mathbf{x}_2), \tilde{\mathbf{y}}_{\text{mix}})$ of a particular attribute mix $\tilde{\mathbf{y}}_{\text{mix}}$, shown below each column (where red denotes 'off' and green denotes 'on').

## 5 Discussion

The results we present generally show there are benefits to mixing. In Table 1 we obtain the best results across SVHN, with $k = 3$ mixup performing the best. ACAI also performed quite competitively, achieving the best results on MNIST and KMNIST. In Table 2 we find that the triplet formulation of mixup (i.e. $k = 3$) performed the best for 20k, 10k, and 5k examples. In Table 3 we experiment with values of $k > 3$ and find that higher-order mixing performs the best amongst our experiments, for instance $k = 8$ mixup for SVHN (256), $k = 3$ mixup for CIFAR10 (256) and $k = 4$ mixup for CIFAR10 (1024). Bernoulli mixup with $k = 2$ tends to be inferior to mixup with $k = 2$, although one can see from Figure 3 that in that regime it generates nowhere near as many possible mixes as mixup, and it would certainly be worth exploring this mixing algorithm for higher values of $k$. While we were not able to achieve ACAI's quoted results for those configurations, our own implementation of it has the benefit of having less confounding factors at play due to it falling under the same experimental setup as our proposed method. Although we have shown that mixing is in general beneficial for improving unsupervised representations, in some cases performance gains are only on

the order of a few percentage points, like in the case of CIFAR10. This may be due to the fact that it is relatively more difficult to generate realistic mixes for 'natural' datasets such as CIFAR10. Even if we took a relatively simpler dataset such as CelebA, it would be much easier to generate mixes if the faces are constrained in pose and orientation than if they were allowed to freely vary (this pose and orientation 'mismatch' be seen in some of the CelebA interpolations in the appendix). Perhaps this would justify mixing in a vector latent space rather than a spatial one. Lastly, in order to further establish the efficacy of these techniques, these should also be evaluated in the context of supervised or semi-supervised learning such as in Verma et al. (2018).

A potential concern we would like to address are more theoretical aspects of the different mixing functions and whether there are any interesting mathematical implications which arise from their use, since it is not entirely clear at this point which mixing function should be used beyond employing a hyperparameter search. Despite Bernoulli mixup not being explored as thoroughly, the disentanglement results in Table 4 appear to favour it, and we also have shown how it can be leveraged to perform class-conditional mixes by leveraging a mixing function to determine what feature maps should be combined from pairs of examples to produce a mix consistent with a particular set of attributes. This could be leveraged as a data augmentation tool to produce examples for less represented classes.

While our work has dealt with mixing on the feature level, there has been some work using mixup-related strategies on the spatial level. For example, 'cutmix' (Yun et al., 2019) proposes a mixing scheme in input space where contiguous spatial regions of one image are combined with regions from another image. Conversely, 'dropblock' (Ghiasi et al., 2018) proposes to drop contiguous spatial regions in feature space. One could however *combine* these two ideas by proposing a new mixing function which mixes spatial regions between pairs of examples in feature space. We believe we have only just scratched the surface in terms of the kinds of mixing functions one can utilise.

One could expand on these results by experimenting with deeper classifiers on top of the bottlenecks, or considering the fully-supervised case by back-propagating these gradients back into the auto-encoder. Note that while the use of mixup to augment supervised learning was done in Verma et al. (2018), in their algorithm artificial examples are created by mixing hidden states *and* their respective labels for a classifier. If our formulation were to be used in the supervised case, no label mixing would be needed since the discriminator is only trying to distinguish between real latent points and mixed ones. Furthermore, if it were to be used in the *semi-supervised* case, any unlabeled examples can simply be used to minimise the unsupervised parts of the network (namely, the reconstruction loss and the adversarial component), without the need to backprop through the linear classifier using pseudo-labels (this would at least avoid the need to devise a schedule to determine at what rate / confidence pseudo-examples should be mixed in with real training examples).

## 6    Conclusion

In conclusion, we present *adversarial mixup resynthesis*, a study in which we explore different ways of combining the representations learned in autoencoders through the use of *mixing functions*. We motivated this technique as a way to address the issue of systematic generalisation, in which we would like a learner to perform well over new and unseen configurations of latent features learned in the training distribution. We examined the performance of these new mixing-induced representations on downstream tasks using linear classifiers and achieved promising results. Our next step is to further quantify performance on downstream tasks on more sophisticated datasets and model architectures.

**Acknowledgments**

We thank Compute Canada for GPU access, and nVidia for donating a DGX-1 used for this research. We also thank Huawei for their support. Vikas Verma was supported by Academy of Finland project 13312683 / Raiko Tapani AT kulut.

## Footnotes

[1]Code provided here: https://github.com/christopher-beckham/amr

* Author is a Canada CIFAR AI Chair

[2]Another way to say this is that for mixing $k$ examples, we sample $\boldsymbol{\alpha}$ from a $k - 1$ simplex. This means that when $k = 2$ we are sampling from a 1-simplex (a line segment), when $k = 3$ we are sampling from a 2-simplex (triangle), and so forth.

[3]Due to time / resource constraints, we were unable to explore Bernoulli mixup as exhaustively as mixup, and therefore we have not shown $k > 3$ results for this algorithm

[4]The architectures we used were based off a public PyTorch reimplementation of ACAI, which may not be exactly the same as the original implemented in TensorFlow. See the anonymized Github link for more details.

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
