[Supplementary Material · AMR_CAMERA_READY_SUPP.pdf]

# 7 Supplementary material

## 7.1 Bernoulli parameters from supervised AMR

To recap, the class mixer in the supervised formulation internally maps from a label $\tilde{\mathbf{y}}_{\text{mix}}$ to Bernoulli parameters $\mathbf{p} \in [0,1]^k$, from which a Bernoulli mask $\mathbf{m} \sim \text{Bernoulli}(\mathbf{p})$ is sampled. This mask is used to construct the mix $\mathbf{mh}_1 + (1-\mathbf{m})\mathbf{h}_2$. The resulting Bernoulli parameters $\mathbf{p}$ are shown in Figure 6, where each row denotes some *discrete* combination of attributes $\tilde{\mathbf{y}}_{\text{mix}} \in \{000, 001, 010, \dots\}$ and the columns denote the index of $\mathbf{p}$ (spread out across four subfigures, such that the first subfigure denotes $\mathbf{p}_{1:128}$, second subfigure $\mathbf{p}_{128:256}$, etc.). We can see that each attribute combination spells out a binary combination of feature maps (purple corresponding to the value 0, and yellow the value 1), which allows one to easily glean which feature maps are representative of a particular set of attributes. We can see that the mask is quite sparse (with most of its values taking on the value 0), and so the resulting mix retains most feature maps from $\mathbf{h}_2$. The mixing function appears to naturally exhibit this behaviour in training, preferring a more 'conservative' approach to producing supervised mixes.

Figure 6: Visualisation of sampled Bernoulli values $\mathbf{m}$ internally produced by the class mixer function. Rows denote discrete attribute combinations $\tilde{\mathbf{y}}_{\text{mix}}$ and columns denote the index of $\mathbf{m}$ (split into four images, such that the first image denotes indices 1 to 128, second image from 129 to 256, and vice versa).

## 7.2 Consistency loss

We experimented with an additional loss on the autoencoder called 'consistency'. This loss was motivated by the fact that some interpolation trajectories we produced did not appear as smooth as we would have liked. To give an example, suppose that we have two images $\mathbf{x}_1$ and $\mathbf{x}_2$, whose latent encodings are $\mathbf{h}_1$ and $\mathbf{h}_2$, respectively. It is not necessarily the case that if one performs a half-way interpolation and decode $g(\frac{\mathbf{h}_1 + \mathbf{h}_2}{2})$ that its re-encoding will also be the same value. In fact, the resulting re-encoding may lean closer to one of the original $\mathbf{h}$'s. This could happen for instance if the two images we are half-way interpolating between are not 'alike' enough, in which case the decoder (in its attempt to fool the discriminator) will simply decode that into a mix which looks more like one of the two constituent images. In order to mitigate this, we simply add another 'reconstruction' loss but of the form $||\tilde{\mathbf{h}}_{\text{mix}} - f(g(\tilde{\mathbf{h}}_{\text{mix}}))||_2$, weighted by coefficient $\beta$.

In order to examine the effect of the consistency loss, we explore a simple two-dimensional spiral dataset, where points along the spiral are deemed to be part of the data distribution and points outside it are not. With the mixup loss enabled and $\lambda = 10$, we try values of $\beta \in \{0, 0.1, 10, 100\}$. After 100 epochs of training, we produce decoded random mixes and plot them over the data distribution, which are shown as orange points (overlaid on top of real samples, shown in blue). This is shown in Figure 12. As we can see, the lower $\beta$ is, the more likely interpolated points will lie within the data manifold (i.e. the spiral).

We also compared $\beta = 0$ and $\beta = 50$ for CelebA, as shown in Figure 13. We can see here that the interpolation trajectory under the latter is smoother, providing evidence that – at least qualitatively – consistency is beneficial to some degree. Unfortunately, due to time constraints we did not explore this in the context of the current iteration of our work (which is primarily focused on measuring how useful the representations are, rather than high quality interpolations *per se*).

Figure 12: Experiments on AMR on the spiral dataset, showing the effect of the consistency loss $\beta$. Decoded interpolations (shown as orange) are overlaid on top of the real data (shown as blue). Interpolations are defined as $||\tilde{\mathbf{h}}_{mix} - f(g(\tilde{\mathbf{h}}_{mix}))||_2$ (where $\tilde{\mathbf{h}}_{mix} = \alpha f(\mathbf{x}_1) + (1-\alpha)f(\mathbf{x}_2)$ and $\alpha \sim U(0,1)$ for randomly sampled $\{\mathbf{x}_1, \mathbf{x}_2\}$)

Figure 13: Interpolations using AMR $\{\lambda = 50, \beta = 50\}$ and $\{\lambda = 50, \beta = 0\}$.

## 7.3 Additional samples

In this section we show additional samples of the unsupervised AMR model (using mixup and Bernoulli mixup variants) on Zappos and CelebA datasets. We compare AMR against linear interpolation in pixel space (pixel), adversarial reconstruction auto-encoder (AE + GAN), and adversarially constrained auto-encoder interpolation (ACAI). Unlike the quantitative results presented in the main paper, our AMR variants here use the aforementioned consistency loss.

- Figure 7: AMR on Zappos (mixup)
- Figure 8: AMR on Zappos (Bernoulli mixup)
- Figure 9: AMR on CelebA (mixup)
- Figure 10: AMR on CelebA (Bernoulli mixup)
- Figure 11: AMR on Zappos-HQ (Bernoulli mixup)

Figure 7: Interpolations between two images using the mixup technique (Equation 4). For each image, from top to bottom, the rows denote: (a) linearly interpolating in pixel space; (b) AE+GAN; (c) ACAI (Berthelot et al., 2019); and (d) AMR ($\lambda = 50, \beta = 50$).

Figure 8: Interpolations between two images using the Bernoulli mixup technique (Equation 5). For each image, from top to bottom, the rows denote: (a) AE+GAN; and (b) AMR ($\lambda = 50, \beta = 50$).

Figure 9: Interpolations between two images using the mixup technique (Equation 4). For each image, from top to bottom, the rows denote: (a) linearly interpolating in pixel space; (b) AE+GAN; (c) ACAI; and (d) AMR ($\lambda = 100, \beta = 50$).

Figure 10: Interpolations between two images using the Bernoulli mixup technique (Equation 5). (For each image, top: AE+GAN, bottom: AMR ($\lambda = 50, \beta = 50$).

Figure 11: Interpolations between two images using the Bernoulli mixup technique (Equation 5). Each row is AMR ($\lambda = 50, \beta = 1$).