[Reviews · NeurIPS 2019]

Reviewer 1



This paper proposes a method for enhancing the latent space learned by auto-encoders, so that the learned latent space produces meaningful features useful for downstream tasks. The proposed approach considers interpolations in the latent space and encourages the reconstructions from these interpolations to be similar to the data using adversarial learning. The learned latent space is shown to capture useful feature via experiments on MNIST, KMNIST and SVHN. The paper presents some promising preliminary experiments. However, there are many issues in the experimental setup Why is the quality of features measured during training ? It is more sensible to learn the features, fix them, and then train the classifier on top of the features. Measuring the classifier accuracy during the training introduces some confounding factors, which makes me question the validity of the results. Why is one method not consistently better than the other ? For instance, in Table 1 mixup(3) is best on MNIST and mixup(2) is best on KMNIST and SVHN. It is unclear to me whether the baselines considered here are comprehensive, significant or strong enough. I appreciate the authors presenting some discussion about this in 211-213. I don’t understand the point of showing the reconstructions from the interpolation (Eg: section 4.3). Many prior work have demonstrated controlling attributes in image generation. Going beyond MNIST, SVHN and evaluating the approach on more complex/real-world datasets would make the paper more compelling. The paper serves as a preliminary exploration on some interesting ideas. However, the experiments need to be performed on real-world datasets against strong baselines with systematic evaluation to demonstrate the benefits of the approach. Questions Is the label based interpolation (section 2.2) used in the quantitative experiments ? Eq (7): Parts of the AC-GAN loss seem to be missing. Other remarks 64: The term ARAE has been used in Zhao et al., Adversarially Regularized Autoencoders. I would suggest using a different term. Make figures 3 and 4 bigger Eq (7) x1, y2 should be x1, y1 ? 151 - 160: I don’t know if the biological motivation needs to be emphasized

Reviewer 2



Significance (6/10) ------------------- Minor insights like the fact that Bernoulli mixup seems to perform worse than linear mixup can be gained from this paper. Based on the results, I don't expect widespread adoption of adversarial mixup as a regularizer but the paper will nevertheless be interesting to some. Originality (5/10) ------------------ The approach seems straightforward based on the ideas that exist in the literature. Quality (6/10) -------------- The quantitative experiments seem reasonable and well executed. What's missing are comparisons with further baselines / better benchmarks. For example, it is not clear from the paper whether I'd want to use any form of adversarial mixing to regularize my classifier compared to or in addition to other data augmentation techniques or dropout, for example. The interpolation results (Figure 1) are surprisingly poor, with not much semantic interpolation and a lot of the kind of ghosting artefacts expected from linear interpolation. The authors claim that a "fundamental difference" between mixup and VAEs is that they "impose no constraint, at least not in the probabilistic sense." I disagree with this statement. In the extreme case of picking each latent dimension from the representation of a different image, you'd be generating independent coefficients just like in a VAE, except you'd sample from an empirical distribution instead of a Gaussian. That is, Bernoulli mixup introduces statistical independence assumptions, even if the authors don't present them as such. It would enhance the paper if the authors could formalize a connection between Bernoulli mixup and nonlinear ICA, and perhaps also explore the probabilistic interpretation of linear mixup. Clarity (7/10) -------------- The paper is mostly well written and clear. Please explain how the parameters of p = embed(y) are trained. Since the image is dependent on y only through the binary mask m ~ p, and this Bernoulli sampling step is non-differentiable, it is not clear to me how this embedding is trained. The authors write that they "collect the highest accuracy on the validation set over the entire course of training". Please make explicit that you used separate validation and test sets to eliminate any doubt that the results are biased. Before Equation 7, the authors write that these losses are optimized "in addition to their unsupervised losses described in Equation 3". The "min_F" in front of the loss in Equation 7 suggests that only these two terms are optimized with respect to F. This should be clearer.

Reviewer 3



This paper proposed a new regularization method to learn better representations through auto-encoders. Intuitively, the proposed method encourages mix in the latent representations so that a combination of latent representations is able to produce realistic images (achieved through an adversarial loss). The high-level idea and formulation is similar to [1], but the proposed method differs in that it is not limited by the specific form of mixing functions, while [1] seems requiring a linear combination of two examples. As also mentioned in the paper, this is a difference between JSGAN and least squares GAN, and the proposed method is able to be freely combined with various mixing functions. In addition to the simplest mixup between two latent representations, this paper proposed and studied two additional mixing functions: (1). an element-wise mask and sum (which is like selection from features maps), and (2). Mixing with K examples (while the experiment only studies K=3). I think the investigation of different mixing functions is original, interesting, and important. The method is mainly evaluated from downstream classification task and qualitative interpolation examples (in the supplementary material). The proposed method outperforms the baselines when the bottleneck dimensionality is 32. Particularly in low data regimes mixing with more than 2 examples plays an important role. However, I think there are several weaknesses of the paper: (1) methodologically, the main difference and advantage of the proposed method over ACAI lie in the flexibility of mixing functions, but in the full training data setting (Table 1) it seems the simplest mixup performs the best. This suggests that it may not be necessary to use different mixing functions. While the proposed method still outperforms ACAI in this case, the gain is more from a different GAN variant under the ACAI framework than a different mixing function, which is less novel and interesting. (2) When the bottleneck dimension is 256, it looks ACAI mixup(3) greatly underperforms ACAI numbers from the original paper. Thus I doubt that the authors’ reimplementation of ACAI is more or less problematic, as mentioned in line 211. Why not include numbers from ACAI mixup(2) from your implementation ? That should be a more fair comparison to the numbers from the original paper to judge whether your re-implementation is correct or not. (3) Given that AMR is much worse than ACAI when dimension is 256 on SVHN dataset, I suspect it might be the same case for other relatively complex datasets. Does this mean the proposed method is inferior when working with high-complexity dataset ? (4) I read all the qualitative examples in the supplementary material, and I cannot tell whether the proposed AMR is better than ACAI or not. [1] Berthelot et al. Understanding and Improving Interpolation in Autoencoders via an Adversarial Regularizer. ICLR 2019 After rebuttal: Authors responded to my points 1-3. Regarding 1, the added CIFAR10 experiments and SVHN with mixup(4) mitigate my concern about the usefulness of varying mixup functions. Regarding 2, it was a miscommunication caused by a typo in the submission and it is now clear. Regarding 3, it seems that AMR only outperforms the authors' re-implementation of ACAI while underperforming the quoted ACAI results on complex settings (e.g. SVHN (256) and CIFAR10 (1024)). I agree that it has less confounders under the same codebase for comparison but I think it is still important to figure out why your implementation fails to reproduce the results. There might be important (but seemingly ignorable) details to the implementation. Overall, I think the authors addressed part of my concerns and I will increase my score accordingly.

[Author Response · NeurIPS 2019]

We thank the reviewers for their time and constructive reviews. We find it encouraging that the ideas we presented were
well-received. The comments from the reviewers were very helpful, and we are eager to use the feedback to further
clarify the paper and add additional results. [Note: mixup(k) / Bern(k) = mixup / Bernoulli mixing $k$ examples at a time]

R1/R3: **Additional experiments**. We ran experiments on CIFAR10 with $d_h = 256$ and $d_h = 1024$. For CIFAR10 (256),
our best result mixup(3) was $0.551 \pm 0.006$ compared to mixup(2) ($0.547 \pm 0.007$) and our baseline ($0.537 \pm 0.004$),
the corresponding quoted result from ACAI is ($0.5277 \pm 0.0045$). For $d_h = 1024$, we get $0.610 \pm 0.009$ for mixup(2)
vs baseline's $0.596 \pm 0.001$ (due to time constraints of this rebuttal, we were unable to let the experiment fully converge
– the quoted ACAI number for this is $0.6399 \pm 0.0047$). We also ran experiments for $k > 3$ on SVHN for $d_h = 256$
and achieved even better results in this regime, e.g. $0.742 \pm 0.021$ for mixup(4) vs $0.653 \pm 0.014$ for mixup(2).

R3: **ACAI implementation fix**. We added in the missing loss term. In short, ACAI outperforms on MNIST/KMNIST,
but on SVHN (both $d_h$ = 32 and 256) and CIFAR10 it does not perform well. We will add these new numbers in.

R2: **Disentanglement metric**. We evaluated the disentanglement scores of our methods on the DSprite (see Beta-VAE
paper) dataset. We found that Bernoulli(3) significantly outperformed ($0.558 \pm 0.01$) the AE baseline ($0.451 \pm 0.027$)
and mixup(3) ($0.511 \pm 0.049$) but not compared to a finely-tuned $\beta$-VAE ($0.652 \pm 0.017$). This gives a stronger
justification for Bernoulli mixup and its utility in the context of disentanglement.

R1: **"Why is the quality of features measured during training ?"** We followed the ACAI paper, which also did this
(i.e see their section 4). Also, as per your suggestion, we will rename ARAE to minimise confusion.

R1: **"Why is one method not consistently better than the other ?"** We performed an analysis comparing the
Lipschitz upper bound (see the 'spectral norm' paper from Miyato et al) of our autoencoder for different values of $k$,
and it appears to increase as $k$ gets larger. This may have implications on our results, and will be explored further.

R2: **"The authors claim that a "fundamental difference" between mixup and VAEs is that they impose no
constraint, at least not in the probabilistic sense." I disagree with this statement."** What we meant was that, unlike
in the VAE setting, we are not explicitly defining a prior function $p(\mathbf{z})$ and enforcing the latent codes to be close to it
(i.e. in a KL divergence sense). We agree in the sense that there is definitely a 'prior' induced with mixup (e.g. the
independence assumptions with Bernoulli), but it is more implicit than what is done in VAEs. We will re-word this.

R2: **"...it is not clear from the paper whether I'd want to use any form of adversarial mixing to regularize my
classifier compared to other..."** Our idea is motivated by a specific problem in generalisation, which is that there may
be certain combinations of latent factors that are poorly represented in the training data. Mixing allows us to explore
these combinations. Furthermore, the paper 'manifold mixup' addresses this concern and shows that mixup behaves
differently to other regularisation schemes, and is competitive with strong baselines (though only uses mixup with
$k = 2$). In our work we focus on a wider class of mixing functions in the unsupervised case. We would like to explore
these in the context of supervised learning but this is beyond the scope of our work. For more evidence that mixing is
desirable – especially in the low data regime, see the 'MixMatch' and 'MixFeat' papers.

R2: **"The interpolation results (Fig 1) are surprisingly poor, with not much semantic interpolation ... ghosting
..."** The supplementary material contains many examples showing semantically meaningful interpolations, on both
CelebA and the Zappos shoe datasets. We compare our results to pixel space interpolation and ARAE, and there is
relatively less ghosting in the AMR interpolations of our approach.

R2: **"Please explain how the parameters of p = embed(y) are trained."** The mask is sampled using the reparameter-
isation trick, which means we are able to backprop through the sampling step and back into the embedding function. Its
parameters are updated in unison with those of the autoencoder. We will update/clarify these equations.

R2: **"Please make explicit that you used separate validation and test sets to eliminate any doubt that the results
are biased."** For each experiment, three different seeds are run, and the best valid accuracy of each seed is taken and
averaged. A val set was only used in our case, though this was an oversight. However, risk of overfitting the val set here
is minimal since we are only training linear classifiers on top of the autoencoder bottleneck (the classification losses *do
not* contribute grads to the autoencoder). If this is concerning, we would be happy to provide held-out test set results.

R3: Addressing **(1)**, our ACAI uses the JSGAN loss, though we also tried LSGAN as per their paper and this did not
appear to make a difference. Assuming mixup with $k = 2$ the only difference is that we don't predict mixing coef, and
contrary to ACAI our generator *also* enforces reconstructions to look realistic by fooling $D$ (which tries to classify them
as fake). Regarding the usefulness of our proposed mixup for $k > 2$, for SVHN (256) and ablations, mixup with $k > 2$
is superior, as well as CIFAR10, and our Bern(3) results on disentanglement. **(2)** typo, this should be 'ACAI mixup(2)'.
For **(3)**, for SVHN $d_h = 256$ only the ACAI results quoted from their paper are superior. While this discrepency should
not be downplayed, our own implementation of it (which is also shown in the table) does not perform as well, and this
has less confounders at play since it is under the same experimental setup as our methods.

[Meta-Review · NeurIPS 2019]

The paper explores the following question: If an autoencoder is learned with adversarial training where the inputs to the discriminator is not the reconstruction from autoencoder but that of a reconstruction using interpolations of pairs (or more) of encodings of the training examples, would that lead to better representation learning? Results on simpler datasets showcases efficacy, while at the same time, evaluating the approach on more complex/real-world datasets would make the paper more compelling. The paper can also benefit from rigorour analysis of the Bernoulli mixup. Aside: crossover in biology happens at recombination hotspots and not at random. They are much more structured.